# Skin Cancer Disease Detection Using Transfer Learning Technique

Javed Rashid [1,2,*], Maryam Ishfaq [3], Ghulam Ali [3], Muhammad R. Saeed [3], Mubasher Hussain [4], Tamim Alkhalifah [5,*], Fahad Alturise [5] and Noor Samand [3]

1   Department of CS&SE, Islamic International University, Islamabad 44000, Pakistan
2   Information Services, University of Okara, Renala Khurd 56130, Pakistan
3   Department of CS, University of Okara, Renala Khurd 56130, Pakistan; maryam.m.ishfaq@gmail.com (M.I.); ghulamali@uo.edu.pk (G.A.); mrrizwan.edu@gmail.com (M.R.S.); mehrveji786@gmail.com (N.S.)
4   MLC Lab, University of Okara, Renala Khurd 56130, Pakistan; mubasher_ashfaq@hotmail.com
5   Department of Computer, College of Science and Arts in Ar Rass, Qassim University, Ar Rass 52571, Qassim, Saudi Arabia; falturise@qu.edu.sa
*   Correspondence: ranajavedrashid@gmail.com (J.R.); tkhliefh@qu.edu.sa (T.A.); Tel.: +92-331-2627786

**Abstract:** Melanoma is a fatal type of skin cancer; the fury spread results in a high fatality rate when the malignancy is not treated at an initial stage. The patients' lives can be saved by accurately detecting skin cancer at an initial stage. A quick and precise diagnosis might help increase the patient's survival rate. It necessitates the development of a computer-assisted diagnostic support system. This research proposes a novel deep transfer learning model for melanoma classification using MobileNetV2. The MobileNetV2 is a deep convolutional neural network that classifies the sample skin lesions as malignant or benign. The performance of the proposed deep learning model is evaluated using the ISIC 2020 dataset. The dataset contains less than 2% malignant samples, raising the class imbalance. Various data augmentation techniques were applied to tackle the class imbalance issue and add diversity to the dataset. The experimental results demonstrate that the proposed deep learning technique outperforms state-of-the-art deep learning techniques in terms of accuracy and computational cost.

**Keywords:** malignant melanoma; deep learning; skin cancer; ISIC-2020 dataset; MobileNetV2

## 1. Introduction

The unchecked increase in irregular skin cells that leads to malignant tumors is skin cancer. Most of these malignancies are caused by unprotected skin exposure to ultraviolet (UV) radiation [1–3]. Melanomas account for 1% of all skin malignancies, with the other 99% being basal cell carcinoma or squamous cell carcinoma [4]. It is one of the most common diseases in American society and a serious one. In the United States alone, more than five million different cases of skin illness are reported every year [5]. For decades, skin cancer has been progressively rising [6]. Melanoma has become the most severe skin cancer and is responsible for around 75% of all skin cancer mortality [7]. The American Cancer Society reported that over 99,780 new cases of melanoma would be discovered in 2022, whereas about 57,100 cases will be reported in men and 42,600 in women. It is expected that about 7650 people will die from melanoma [4]. Melanoma affects the melanocytes (squamons cell layer). Based on cancerous cell severity, it may be further divided into benign and malignant categories. A benign skin lesion is a mole or tag that does not contain cancerous cells. Malignant lesions necessitate immediate treatment due to a high concentration of cancer cells [8]. According to current figures, the survival rate is 99% if the melanoma is detected before spreading near lymph nodes [4]. The survival rate is about 68% after melanoma spreads near lymph nodes, and the survival rate is about 30% in case the melanoma spreads near lymph nodes and other organs [4]. The statistics show that in

2019 about 1,361,282 people were living with melanoma [9]. In 2020, about 324,635 people were diagnosed as melanoma patients, and about 57,043 died from melanoma [10].

Doctors use a variety of ways to detect skin cancer. An expert dermatologist usually follows a series of benchmarks, starting with naked-eye recognition of suspicious tumors, then dermoscopy, and finally a biopsy [11,12]. It can take a long time, and the person may advance to a later step. The detection performance of dermoscopic images has increased by 50%, with absolute accuracy ranging from 75% to 84% [13]. Furthermore, correct diagnosis is unique and highly dependent on the clinician's abilities [14]. The manual identification of skin diseases is very tough and tiring for patients [15]. Because computer-assisted diagnosis helps the medical experts in analyzing the dermoscopy procedures in case of a lack of expertise in the diagnostic process and lack of availability of a professional [16,17]. A computer-based classification is an option for diminishing inter- and intra-variability. The state-of-the-art computer-assisted dermatological image categorization systems had two fundamental flaws; there are inadequate data [18], and the imaging process is the second difficult challenge, in which skin images are obtained using a specific instrument called dermoscopy [19], whereas other medical images, such as biopsy images and histology images, are obtained using biopsy and microscope. The state-of-the-art approaches [20] needed substantial preprocessing, segmentation, and feature extraction operations to categorize skin images.

Artificial intelligence is a novel area; the revolution related to it is similar to that made by adding techniques to every part of our lives [21–23]. Machine learning (ML) methods assist in avoiding the step of manually extracting features and help perform classification tasks [24] efficiently. Recently, there has been growing attention in employing ML approaches to help accurate cancer detection [24,25]. Machine learning algorithms have significantly increased cancer prediction accuracy by 15% to 20% during the last few decades [25]. Deep learning [26–30] is one of AI's most rapidly growing topics due to its broad range of applications. Deep learning, specifically convolutional neural networks (CNNs) powered by sophisticated computer techniques and massive datasets, has become one of the most potent and popular ML approaches in image identification and classification [31] and has been used to categorize skin lesions [32,33]. The preliminary information and complex image preprocessing methods required for image classification using traditional ML methods are no longer in demand. Some deep-learning-based classifiers have demonstrated the ability to classify skin cancer images with the same accuracy as dermatologists [33]. As a result, CNNs can assist in developing computer-aided rapid skin lesion classifiers at the level of dermatologists.

However, high-quality medical imaging datasets for training are still scarce. It is predominantly related to the absence of annotated/labeled images for abnormal classes [34]. CNN with simple architecture is more likely to overfit on limited training datasets. Some researchers use extremely deep CNNs models (e.g., Resnet152 contains 152 layers) [35]). Although this improves network classification performance and increases computing costs, that is a major problem for clinical applications [36,37]. Moreover, researchers are using pre-trained CNNs to classify skin lesions [38–42], which prevents the issue of overfitting, and pre-trained CNNs use features learned from real-world image datasets (such as ImageNet).

The present study proposes a deep transfer learning technique for melanoma classification based on MobileNetV2. For melanoma detection and recognition, pre-processing and heavy augmentation methods are used for the first level to overcome the imbalanced class problem in the ISIC-2020 challenge dataset. In the second stage, the transfer learning MobileNetV2 architecture is used for auto feature extraction and classification as benign or malignant.

The remaining of the article is organized as follows. In Section 2, a detailed related work of the existing approaches is discussed. The materials and methods are discussed in Section 3. In Section 4, the results and discussion are presented. Finally, Section 5 describes the conclusion and future work, followed by the references.

## 2. Related Work

Various techniques have been proposed for melanoma classification in the previous few decennaries. Most methods [43–45] used image processing techniques to extract features and then fed them into a classification technique. Khan et al. [46] presented a detection and classification technique between melanoma and nevi. At first, the author applied a Gaussian filter to remove noise. K-mean clustering was used for lesion segmentation. Then, textural and color features were extracted using a hybrid super feature vector. After that, support vector machines (SVMs) were applied for classification. The proposed methodology obtained 96% accuracy on the ERMIS dataset. Filali et al. [47] presented a new technique based on combining deep learning (DL) and handcrafted features. The developed method obtained 87.8% on the ISIC challenge dataset and 98% accuracy on the Ph2 dataset. Hu et al. [48] used an approach based on feature similarity measurement, and then SVM was used for classification. Abbas et al. [49] presented a five-layer system known as "DermoDeep" to differentiate between nevi and melanoma. This method integrated visual features and a five-layer model to achieve the best classification results. Dalila et al. [8] extracted three types of features (texture, geometrical properties, and color) and selected optimal features using ant-colony-based segmentation. Then ANN was used for classification. Almansour et al. in [50] proposed an approach in which textual features were extracted, and then SVM was implemented as a classifier. The presented model achieved 90% accuracy on 227 images. Pham et al. [51] used image enhancement techniques for extracting ROIs. After that, SVM was used for the classification of the pre-processed images. The attained accuracy was 87.2%. Yu et al. [52] introduced a method to enhance the images for extracting ROIs and used a deep residual model to classify images. The obtained accuracy of the proposed system was 85.5%.

Recently, researchers have been working on melanoma classification by using deep learning models. Yu et al. [53] developed a new method depending on deep CNN and feature encoding techniques (FV encoding) to create more meaningful features for accurate melanoma recognition. The developed model was trained on the ISIC 2016 dataset and archived with a 86.54% accuracy. Rokhana et al. [54] proposed a deep CNN architecture to classify melanoma dermoscopy images into benign skin lesions and malignant melanoma. The presented approach was evaluated on the ISIC-archive repository. The proposed approach gained 91.97% sensitivity, 84.76% accuracy, and 78.71% specificity. Xie et al. [7] used a classification method based on the ensemble model. Liberman et al. [55] developed an ensemble model based on three classifiers to classify mole images in non-melanomas and melanomas. Zhou et al. [56] presented a new method based on spiking neural networks with time-dependent spike plasticity. Hosny et al. [57] implemented a deep CNN architecture for melanoma classification. The presented method was tested on three different datasets. Mukherjee et al. [58] used a CNN-based method known as CNN malignant lesions detection (CMLD). The developed model achieved 90.14% and 90.58% accuracy on MED-NODE and Dermofit datasets. Esteva et al. [59] presented a technique for detecting skin diseases as an initial stage and classifying skin cancer using deep networks.

Cakmak et al. [60] presented a deep neural network-based model called Nasnet Mobile to detect melanoma. The presented technique was evaluated on the HAM10000 dataset. Various augmentation techniques were used to tackle the problem of imbalanced classes. The proposed model obtained the accuracy with the Nasnet-Mobile network was 89.20% without data augmentation and 97.90% with data augmentation. Brinker et al. [61] used a pre-trained architecture named ResNet50 to classify the skin lesion as melanoma or nevi. The proposed model achieved 77.9% and 82.3% ratios for sensitivity and specificity, respectively. Han et al. [62] utilized the ResNet152 model to classify various skin lesions. The specificity and mean sensitivity for three different lesions, melanoma, seborrheic keratosis, and nevi, were 87.63% and 88.2%, respectively. Hosny et al. [63] replaced the last three layers of AlexNet with fully connected layers, softmax, and an output layer to classify skin lesions. The proposed algorithm achieved 96.86% accuracy. Esteva et al. [64] used a pre-trained model named Inception-v3 to classify skin lesions. They increased the testing

dataset by using augmentation techniques. The proposed classification model obtained 71.2% accuracy. The summary of the related work is presented in Table 1.

**Table 1.** Related work summary.

| Reference | Methodology | Disease | Dataset | Accuracy |
|---|---|---|---|---|
| [60] | Nasnet Mobile with Transfer Learning | Melanoma | HAM10000 skin lesion dataset | 97.90% |
| [46] | Support Vector Machine (SVM) | Melanoma, Nevus | DERMIS dataset | 96.0% |
| [53] | DCNN-FV | Melanoma, Non-Melanoma | ISBI 2016 challenge | 86.54% |
| [54] | Deep Convolutional Neural Network (CNN) | Benign, Malignant, Melanoma | ISIC Archive Repository | 84.76% |
| [47] | SVM | Melanoma, Non-Melanoma | Ph2 & ISIC Challenge | Ph2 98%, ISIC 87.8% |
| [7] | Ensemble Model | Malignant, Benign | Xanthous Race (XR), Caucasian Race (CR) | XR (94.14%), CR (91.11%) |
| [48] | FSM & SVM | Malignant, Benign | Ph2 | 91.90% |
| [49] | DCNN | Melanoma, Nevi | Self Contained 2800 Images | 96% |
| [8] | ANN | Malignant, Benign | Self Contained 172 Images | 93.6% |
| [61] | ResNet-50 | Melanoma, Nevi | Self Contained 4204 Images | Sensitivity (82.3%), Specificity (77.9%) |
| [55] | Ensemble Model | Melanoma, Non-Melanoma | ISIC | Avg Precision (98.0%) |
| [56] | STDP based Spiking NN | Malignant, Melanoma, Benign, Nevi | ISIC 2018 | 87.7% |
| [57] | DCNN | Melanocytic, Non-melanocytic | MED-NODE DermIS & DermQuest (D&D), ISIC-2017 | MED-NODE (99.29%), D&D(99.15%), ISIC(98.14%) |
| [58] | CNN based CMLD model | Melanoma, Benign | Dermofit, MED-NODE | Dermofit (90.58%), MED-NODE (90.14%) |

## 3. Materials and Methods

Self-learning algorithms are the foundation of artificial intelligence. As new information about the work is received, such algorithms continue to evolve [65]. These techniques are continuously evolving to resolve these issues. Self-learning algorithms can function because these models are based on the human brain [66]. Artificial neural networks (ANNs) are nodes (neurons) connected at various levels, such as human nerve cells. Information is recorded, processed (via positive or negative weighting), and output inside this neuron network. ANNs look especially promising because they have many levels and can recognize more complicated patterns. Deep learning [67,68] refers to the learning processes that such networks can perform.

This research introduces a deep transfer learning system to classify melanoma skin cancer. At the first level, pre-processing and various augmentation approaches are used to resolve the issue of class imbalance in the dataset and generate diversity. At the second level,

auto features are extracted, and then a pre-trained "MobileNetV2" model is implemented to classify the malignant melanoma from a benign skin lesion. The flow chart of the proposed technique is presented in Figure 1.

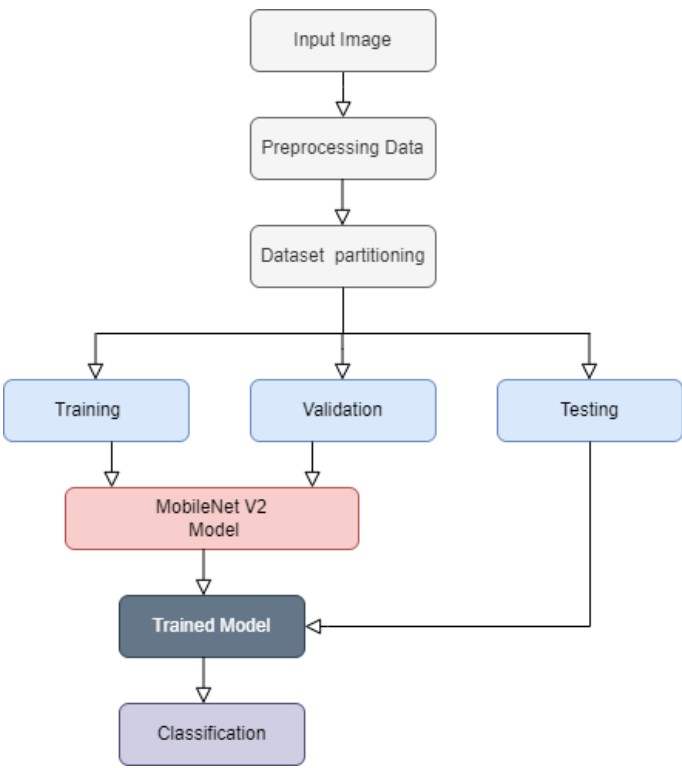

**Figure 1.** Proposed method flowchart.

### *3.1. Dataset*

The performance of the deep learning techniques is based on the availability of a suitable and valid dataset. The following dataset is being used in this research.

### 3.1.1. SIIM-ISIC 2020 Dataset

The ISIC-2020 Archive [69] comprises the world's most enormous number of quality-controlled skin lesions dermoscopic images publicly available for research. Several institutions contributed data from patients of various ages and sexual orientations. The dataset includes 33,126 dermoscopic images, 584 images related to malignant, and 32,542 benign skin lesions from more than 2000 patients. Every image is associated with one of these patients through a unique patient identifier. We used 11,670 images of benign class and 584 images of melanoma. Considering the data of these two classes is imbalanced. Therefore, to handle the class imbalance issue, 4522 melanoma images were included in the ISIC 2019 archive [70]. After that, various data augmentation strategies were performed, including rescaling, width shift, rotation, shear range, horizontal flip, and channel shift, which became 11,670 after augmentation. The reason behind using 11,670 images of benign is to tackle the class imbalance issue. The images of the benign class were selected arbitrarily from the whole set of images. See sample images in Figure 2 and details of the classes in Table 2.

**Table 2.** Summary of the ISIC-2020 dataset.

| Class Labels | Training | Validation | Testing |
|---|---|---|---|
| Melanoma | 8170 | 1750 | 1750 |
| Benign | 8170 | 1750 | 1750 |
| Total | 16,340 | 3500 | 3500 |

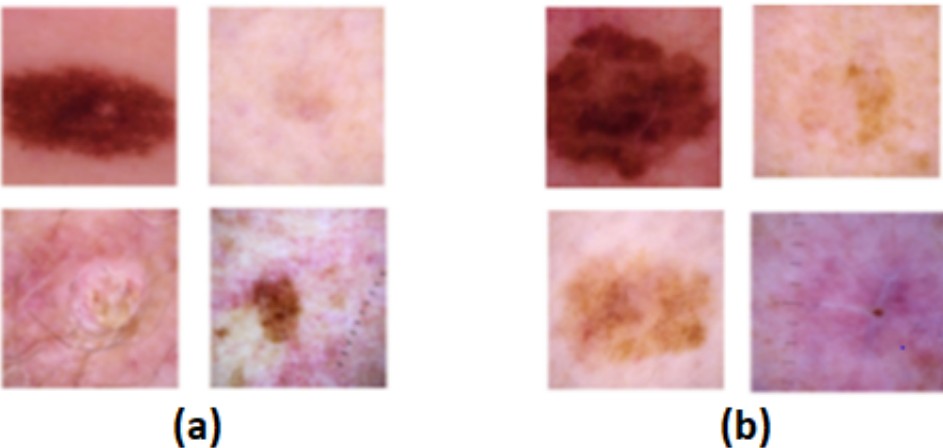

**Figure 2.** (**a**) Benign and (**b**) melanoma lesions images.

*3.2. Image Pre-Processing*

To obtain higher consistency in classification results and improved features, preprocessing is employed for all input images of the ISIC-2020. The CNN approach requires a massive amount of repetitive training; for this purpose, a large-scale image dataset was required to prevent the danger of over-fitting.

### 3.2.1. Image Resizing

All images in the original ISIC dataset are available in $6000 \times 4000$ dimensions. The dataset is resized to $256 \times 256$. It will reduce the model performance dramatically and speed up the processing process.

### 3.2.2. Data Augmentation

Various data augmentation approaches have been applied to the training set with the help of the image data generator function of the Keras library in Python to overcome overfitting and increase the dataset's diversity. The computational cost was decreased by utilizing smaller pixel values within the same range; this was accomplished using scale transformation. Therefore, the value of each pixel ranged from 0 to 1 with the help of the parameter value (1. /255). The rotation transformation was used to rotate the images to a particular angle; therefore, 25° was used to rotate the images. Images can be shifted arbitrarily to the right or left by employing the width shift range transformation; the width shift parameter was set to 0.1. With a value of 0.1, the height shift range parameter was used to move the training images vertically. Shear transformation is a technique in which one axis of an image is fixed, and then the other axis is stretched to a certain angle called a shear angle; in this case, a 0.2 shear angle was used. The zoom range argument was used to perform the random zoom transformation; a value greater than 1.0 implies that the images were magnified, and a value less than 1.0 means that the images were zoomed out. As a result, a zoom range of 0.2 was used to magnify the image. Flip was used to flipping the picture horizontally. Brightness transformation was used, in which 0.0 represents no brightness and 1.0 represents maximum brightness; as a result, the zoom range 0.5–1.0 was used. In channel shift transformation, the channel values are randomly shifted by a random value chosen from the particular range; as a result, the 0.05 channel shift range was applied, and the fill mode was the closest, as shown in Table 3.

**Table 3.** Image augmentation techniques.

| Transformations | Setting |
|---|---|
| Scale transformation | ranged from 0 to 1 |
| Rotation transformation | 25° |
| Zoom transformation | 0.2 |
| Horizontal flip | True |
| Shear transformation | 20° |

### 3.3. Training, Validation and Testing

The ISIC-2020 dataset was composed of three portions: training, testing, and validation. The training set was utilized for training the MobileNetV2 model, and the validation and test datasets were used to evaluate the performance of the introduced model. Therefore, we split the dataset into training, testing, and validation, with 70%, 15%, and 15%, respectively. The MobileNetV2 model was trained using the dataset presented in Section 3.1.1. For the ISIC-2020 dataset training, validation, and testing, 16,350, 3500, and 3500 images were used.

### 3.4. MobileNetV2 Architecture

In the current study, deep transfer learning MobileNetV2 [71] architecture is to tackle the issue of melanoma classification. Several different factors influenced the selection of the MobileNetV2 model. The dataset used for training a model was relatively tiny, making it susceptible to over-fitting, and using a small but more expressive system, like MobileNetV2, mitigated this effect significantly. MobileNetV2 is a framework that optimizes execution speed and memory consumption at a minimal cost with respect to the error [71]. Due to the high execution speed, parameter tuning and experimenting are considerably more manageable, while minimal memory consumption is an additional attractive feature. The main structure of MobileNetV2 is based on its previous version, MobileNetV1. Two significant notions explaining the MobileNetV2 framework are the depthwise separable convolution, linear bottleneck, and the inverted residual, which are discussed further.

#### 3.4.1. Depthwise Separable Convolutions

As discussed in [71], other efficient networks, such as ShuffleNet [72] and Xception [73], utilize the depthwise separable convolution. The Depthwise separable convolution used in MobileNetV1 was also applied in MobileNetV2 [74]. Depth-wise separable convolution replaces traditional convolution with two procedures. The first procedure is a features map-wise convolution, which means a different convolution is applied to each feature map. The feature maps that come from this process are stacked, and the second procedure, a pointwise convolution, is used for these feature maps to process. In this case, the pointwise convolution is implemented with a $1 \times 1$ kernel and is implemented to every feature map at once. The image is processed simultaneously in height, width, and channel dimensions in a traditional convolution, as shown in Figure 3.

However, the depthwise separable convolution analyzes the image by height and width dimensions during the first procedure. It handles the channel dimensions during the second procedure, which refers to a factorization of the traditional convolution.

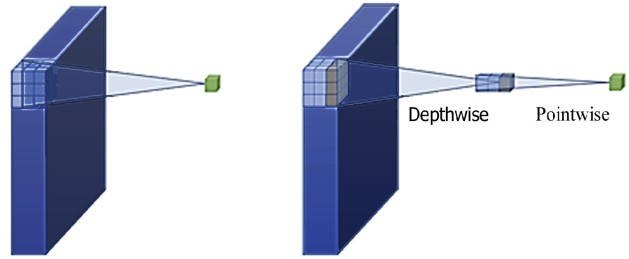

**Figure 3.** Traditional convolution and depth-wise separable convolution.

### 3.4.2. Linear Bottleneck and Inverted Residual

In [71], the inverted residuals were explained and compared with residual blocks [35], which are an integral part of the ResNet network. Both blocks make use of bottleneck and residual connections, and both utilize three convolutional operators. The first and last operators make use of 1 × 1 filters [35,71], which translate data from the input domain to an intermediary representation and from the intermediary representation to the outcome domain. A three-by-three (3 × 3) filter [35,71] is used to process the intermediate representation, as shown in Figure 4.

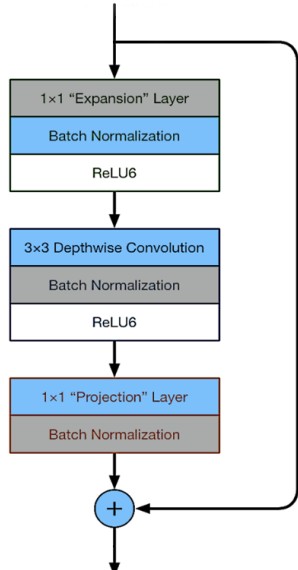

**Figure 4.** Bottleneck residual block.

The initial and final residual block convolutions have a greater number of feature maps than the block inner convolution [35]; on contrary, the inverted residual employs the first and final convolutions with a lesser number of feature mappings than the inner convolution [71]. In both situations, the residual link is between the initial and final feature maps (channels), which are fewer in the scenario of MobileNetV2 as compared to ResNet [35]. When multiple units are stacked together in either architecture, the outcome is an alternation of small and big layer results. The memory efficiency is achieved using the residual block arrangements of the MobileNetV2 [71].

MobileNet V2 includes an expansion layer of 1 × 1, depth-wise and pointwise convolutional layers in each block. In contrast to V1, MobileNetV2 contains pointwise convolutional layers termed the projection layer, which transforms data with many channels into a tensor with a significantly smaller number of feature maps (channels). The bottleneck residual block, which contains the outcome of every block, is a bottleneck in the system. An expansion convolutional layer of 1 × 1 will increase the number of feature maps (channels) based on the expansion factor before passing through the depth-wise convolution. The residual connection is the second new feature introduced in MobileNetV2's core component. A residual connection is established to facilitate gradient flow through the system. Every layer of the MobileNetV2 architecture includes batch normalization, with the ReLU6 as the activation function. The outcome of the projection layer, on the other hand, does not contain an activation function. The whole MobileNet V2 structure is comprised of 17 bottleneck residual blocks in a queue, followed by a 1 × 1 regular convolution, a global average pooling layer, and then a classification layer. The pre-trained MobileNetV2 is shown in Figure 5. Table 4 shows the model and parameters that produced the best results, with an accuracy of 98.2 percent.

**Table 4.** Parameters used in the Experiment

| Parameters | Values |
| --- | --- |
| Architecture Used | MobileNetV2 |
| Type of Transfer | From scratch transfer Knowledge |
| Train Layers | All |
| Learning Algorithm | Adam |
| Learning rate | Default Alpha Rate |
| Activation Function | ReLu & Sigmoid |
| Loss Function | binary-cross-entropy |
| Batch Size | 64 |
| Epochs | 100 |

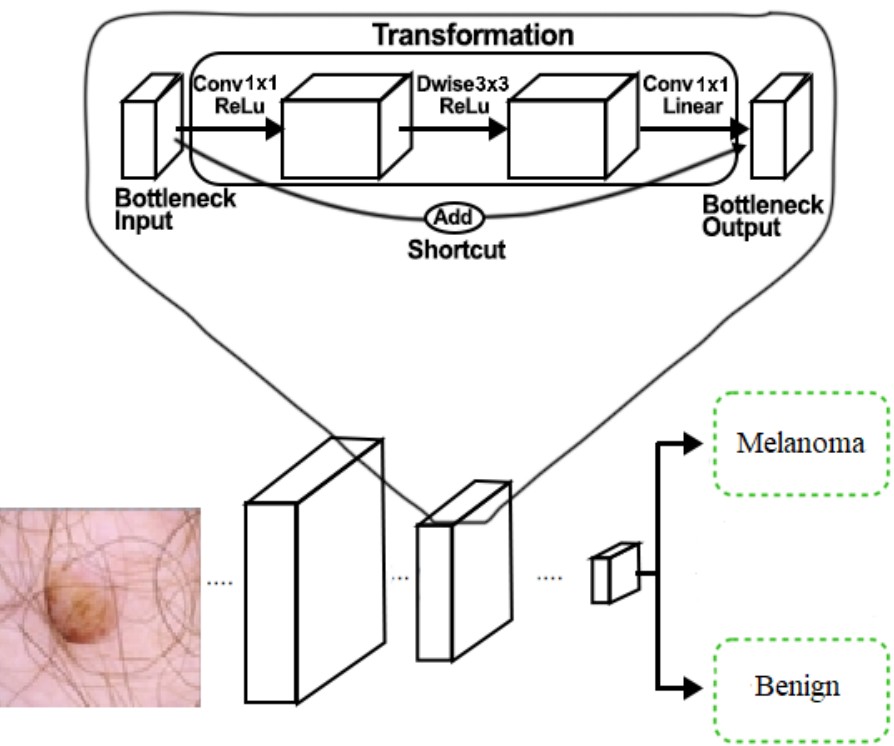

**Figure 5.** Classifier based on MobileNetV2.

### 3.5. Evaluation Measures for Classification

After the training process, the proposed technique was tested on the testing dataset. The architecture's performance was validated using the accuracy, F1 score, precision, and recall. The performance metrics employed in this research are explored in detail below. The definitions and equations are mentioned below, where TP stands for true positives, TN stands for true negatives, FN stands for false negatives, and FP stands for false positives.

#### 3.5.1. Classification Accuracy

The classification accuracy is measured as the percentage of correct predictions to the total number of accurate predictions.

$$\text{Accuracy} = \frac{TP + TN}{(TP + TN + FP + FN)} \tag{1}$$

### 3.5.2. Precision

Several examples demonstrate that classification accuracy is not always a valid metric for overall model performance. One of these cases is when the distribution of classes is imbalanced. If we treat all samples as being of the highest quality, we will obtain a high accuracy rate, which makes no sense. On the other hand, precision indicates that inconsistency can be found when repeatedly utilizing the same instrument, for instance, when measuring the same part. Precision is one of such measures, which is characterized as

$$\text{Precision} = \frac{\text{TP}}{(\text{TP} + \text{FP})} \tag{2}$$

### 3.5.3. Recall

A recall is another vital statistic, which can be defined as dividing input samples into classes that are successfully predicted by the system. The recall is calculated as

$$\text{Recall} = \frac{\text{TP}}{(\text{TP} + \text{FN})} \tag{3}$$

### 3.5.4. F1 Score

The f1 score is a well-known metric that measures precision and recall in a single metric. The f1 score is calculated as

$$\text{F1 Score} = \frac{2 * (\text{Precision} * \text{Recall}}{(\text{Precision} + \text{Recall})} \tag{4}$$

### 3.5.5. AUC Score and ROC Curve

The area under curves (AUC) reflects the level of separability, and the receiver operating characteristic (ROC) is a probability curve. The ROC curve is a graph that depicts the connection between specificity (rate of false positives) and sensitivity (true positive rate).

## 4. Results and Discussion

The experiment with the presented MobileNetV2 architecture was carried out on Google Colab. The MobileNetV2 technique was implemented on the Tensor-Flow platform, open-source Keras packages, and the Python programming language. For training, it used the Adam optimizer with a default learning rate and a binary cross-entropy loss function. The results of the proposed MobileNetV2 model focused on the following:

1.  To differentiate the dermoscopic images into malignant or benign.
2.  Evaluated the performance of the presented MobileNetV2 model on the ISIC-2020 dataset by using various data augmentation techniques.
3.  The results were compared with state-of-the-art techniques.

### 4.1. Proposed Model Performance on ISIC-2020 Dataset

The experiment was conducted to assess the performance of the introduced MobileNetV2 architecture. In the experiment, the Adam optimizer, binary-cross-entropy loss function, 100 epochs, 64 batch size, and the default alpha rate were used as shown in Table 4. The experimental outcomes presented that the introduced method obtained 98.1% and 98.4% accuracy for melanoma and benign lesions, respectively. It also obtained 98.2% average accuracy on the ISIC-2020 dataset, as presented in Table 5. The transfer learning model achieved 98.3% and 98.0% recall on melanoma and benign skin cancer. It obtained a 98.1% F1-score on both diseases, and 98.0%, 98.3% precision on melanoma and benign diseases, respectively. The ISIC 2020 test (leader board) results showed that the proposed method obtained 98.04%, which means there was not enough of a difference between the two test accuracies, our test set and the leader board test set, as shown in Table 5. There were 10,982 images in the ISIC 2020 test set on the leader board, with 690 unique patient IDs

and 10,292 duplicates. Because the ground truth for ISIC 2020 was not publicly available, the organizer's statistic on Kaggle area under the receiver operating characteristics curve (AUC) was used.

The accuracy and losses in every epoch during training and validation are shown in Figure 6. It shows that after 10 epochs, the training and validation accuracies increased rapidly, and it was steady after almost 40 epochs. On the other hand, training and validation losses decreased rapidly after 10 epochs, and after 60 epochs, the losses of training and validation became stable. The results demonstrated that the proposed method yielded higher classification scores on the ISIC-2020 dataset when the data augmentation strategies were used in the training set.

**Table 5.** Classification accuracies, recall, precision and F1-score of presented MobileNetV2 Model on ISIC-2020 dataset.

| Performance Measure | Melanoma | Benign | Average Accuracy | Leader Board Accuracy |
|---|---|---|---|---|
| Accuracy | 98.1% | 98.4% | 98.2% | 98.04% |
| Recall | 98.3% | 98.0% | - | - |
| F1-Score | 98.1% | 98.1% | - | - |
| Precision | 98.0% | 98.3% | - | - |

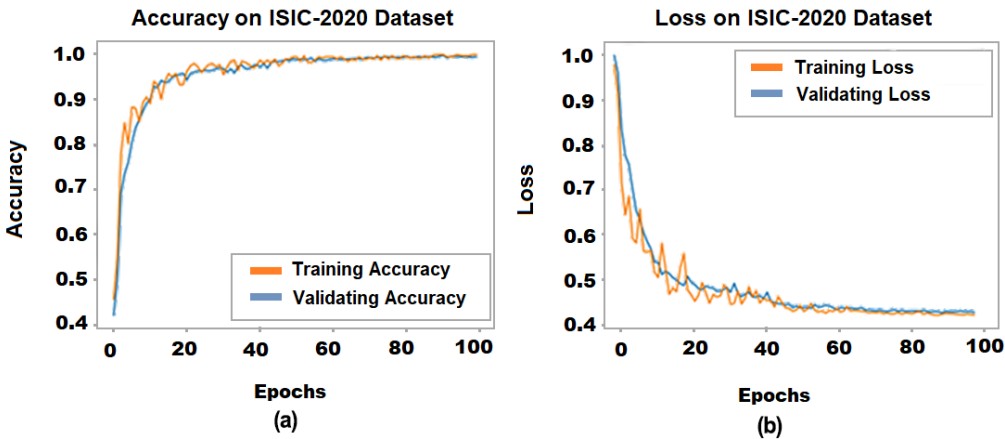

**Figure 6.** (**a**) Accuracies graph, and (**b**) loss graph of the proposed model.

The confusion matrix is a valuable ML method that determines the recall, accuracy, ROC curve, and precision of a model. A confusion matrix was used to measure the classification accuracy visually. It indicated the greater classification accuracy of the MobileNetV2 of the appropriate class in a dark color, whereas a lighter color indicated the incorrectly identified samples. Correct predictions were displayed diagonally in the confusion matrix, whereas incorrect predictions were displayed off-diagonally in the confusion matrix. As demonstrated by the results, the presented MobileNetV2 framework outperformed when data augmentation methods were implemented in the ISIC-2020 dataset, as indicated in Figure 7. It indicated that the MobileNetV2 model correctly identified 1721 benign lesion images out of 1750 and 1556 malignant images out of 1750. The overall accuracy of the presented MobileNetV2 system was 98.2%, and 1.8% error, which indicated the introduced MobileNetV2 model's generalization.

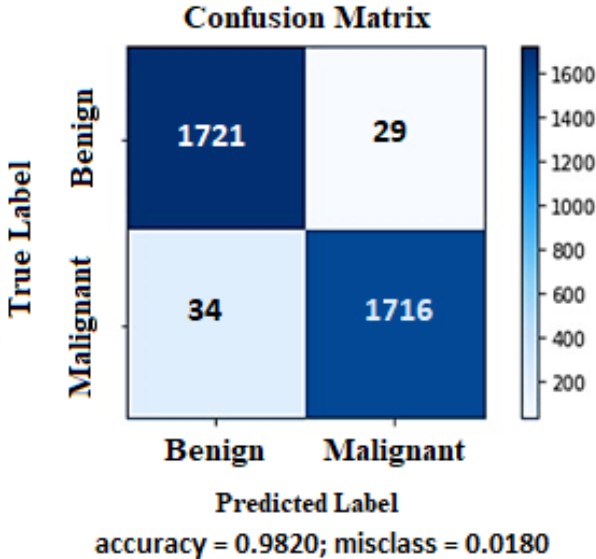

**Figure 7.** Confusion matrix of the MobileNetV2 model.

The MobileNetV2 model demonstrated outstanding classification performance in validation and test set classes by having a larger area under the curve (almost 98.2%). The developed methodology's performance was measured using the ROC curve as depicted in Figure 8. The black indicated the ROC curve, and the red indicated the random guessing.

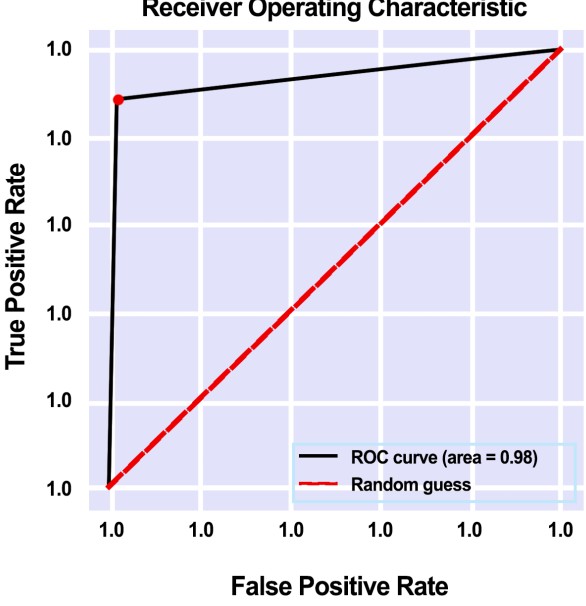

**Figure 8.** MobileNetV2 ROC curve on ISIC-2020.

The evaluation metrics, including accuracy, F1-score, recall, precision, and the ROC curve, demonstrated that the proposed method performed exceptionally well on the ISIC-2020 dataset when the data augmentation strategies were used in the training set.

### 4.2. Comparison with State-of-the-Art Methods

To represent the generalization of the introduced approach, we compared the performance of the presented model with state-of-the-art techniques. It was observed that the presented deep learning system outperformed state-of-the-art approaches. There was a slight variation in misclassification when comparing the proposed strategy to state-of-the-art methods. The developed method's performance was evaluated compared to other

melanoma and benign classification strategies that were earlier published. The study results revealed that the introduced method had the highest accuracy compared to other current research, as indicated in Table 6.

The proposed MobileNetV2 model outperformed the existing studies, as Mukherjee et al. [58] obtained 90.58% accuracy in classifying the melanoma and benign skin cancer diseases on the Dermofit dataset and 90.14% accuracy in the MED-NODE dataset using the CNN-based CMLD model. Dalila et al. [8] reported 93.6% accuracy using the ANN-based model on a self-created dataset. It used only 172 images to classify the melanoma and benign diseases. Hu et al. [48] used FSM and SVM models performed 91.9% accuracy and used the Ph2 dataset. The model presented in [7] can classify the melanoma and benign skin diseases with 94.14% on XR dataset and 91.11% accuracy on the CR dataset. Another research conducted by Mijwil [75] obtained 86.90% accuracy using ISIC2019 and ISIC2020 datasets to distinguish between melanoma and benign diseases. We can say that the proposed MobileNetV2 model dominated the existing techniques and thus achieved 98.2% accuracy. It obtained the highest accuracy compared to existing models, as shown in Table 6.

**Table 6.** Comparison with state-of-the-art models.

| Ref. | Methodology | Diseases | Dataset | Accuracy |
| --- | --- | --- | --- | --- |
| [58] | CNN based CMLD model | Melanoma, Benign | Dermofit, MED-NODE | Dermofit (90.58%), MED-NODE (90.14%) |
| [8] | ANN | Melanoma, Benign | Self Contained 172 Images | 93.6% |
| [48] | FSM & SVM | Melanoma, Benign | Ph2 | 91.90% |
| [7] | Ensemble Model | Melanoma, Benign | Xanthous Race (XR), Caucasian Race (CR) | XR (94.14%), CR (91.11%) |
| [75] | InceptionV3, ResNet, and VGG19 | Melanoma, Benign | ISIC archive between 2019 and 2020 | 86.90% |
| | Proposed Method | Melanoma, Benign | ISIC2020 | 98.20% |

## 5. Conclusions and Future Work

Melanoma is the worst type of skin cancer, but if caught in a timely enough manner, it can be a non-life-threatening disease. As a result, it is critical to employ supportive imaging modalities that have been proved to help with diagnosis. These methods are based on procedures devised by doctors to detect melanoma before it spreads to nearby lymph nodes. In this research, we provide a transfer learning model for melanoma and benign skin lesions diagnosis based on MobileNetV2, which can be used to investigate any suspicious lesion. The suggested method is applied to an ISIC2020 challenge dataset of skin cancer disorder images to determine if a disease is malignant or benign. Data augmentation techniques were used to increase the dataset's size and improve the accuracy of MobileNetV2. This architecture works effectively and has a diagnostic accuracy of 98.2 percent. Finally, the accuracy of various state-of-the-art models is compared to the proposed framework. The suggested architecture was found to provide outstanding classification accuracy without needing model training from scratch to improve model efficiency. After a sufficient number of high-resolution photographs is acquired, this study will be carried out on a series of skin cancer images for patients from Pakistan in the future.

**Author Contributions:** The research conception, technique, and programming were proposed by J.R., G.A., M.R.S. and G.A. are in charge of the technical and theoretical framework. T.A., F.A. and others created the datasets. M.I., M.H. and N.S. were in charge of the technical review and improvement. G.A. and J.R. provide comprehensive technical assistance, direction, and supervision. M.I. and J.R.

are in charge of editing and proofreading. All authors have read and agreed to the published version of the manuscript.

**Acknowledgments:** The researchers would like to thank the Deanship of Scientific Research, Qassim University for funding the publication of this project.

**Funding:** This research received no external funding.

**Institutional Review Board Statement:** Not applicable.

**Informed Consent Statement:** Not applicable.

**Data Availability Statement:** Not applicable.

**Conflicts of Interest:** The authors declare no conflict of interest.

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
