# Peer review of "Skin Cancer Disease Detection Using Transfer Learning Technique"

_applsci, doi:10.3390/app12115714_

Round 1
Reviewer 1 Report
Incorrect formatting title page isn’t correct, and you have unfilled sections, e.g. 6. Patents
Spelling needs to be checked thoroughly, e.g. Figure 6
Training image counts adds up to 16340 not 16350
Dataset numbers are very odd you miss out a large number of images without any reasoning
It isn’t clear but appear no comparison against other networks is performed. The first table seems to just be paper authors scores.
You say you split and dive the dataset, but you state in section 3.1 you use all ISIC 2020 dataset so the network has learned from the training and validations set.
You also need to demonstrate your results on the ISIC leader board to demonstrate your method over others. Not just on your testing set.
The paper need a lot of correcting and checking over and fair comparisons. Statements need to be doubled checked as many do not makes sense or fit in the section.
Figures need standardising and to be in-house not from google.
Author Response
We are thankful to the reviewers for appreciating the strength of our article. We are also thankful for their valuable comments and suggestions which helped us to improve the manuscript (ID: applsci-1713218) entitled “Skin Cancer Disease Detection using Transfer Learning Technique”. The revised version has been prepared to address the reviewers’ suggestions. We have tried our best to answer their questions here and also included the relevant information in the revised paper as well. The author’s responses to the reviewers’ comments are highlighted here in YELLOW color whereas the actual modifications necessary to be made in the paper are accentuated in YELLOW. Detailed responses to the reviewers’ comments are given below.
Reviewer 1:
Incorrect formatting title page isn’t correct, and you have unfilled sections, e.g. 6. Patents
Response: Dear Reviewer, thank you for your valuable comment. The formatting if title corrected and title page has been corrected and patents section has been removed successfully.
Spelling needs to be checked thoroughly, e.g. Figure 6
Dear Reviewer, The text in Fig. 6 has been corrected, and also speeling mistakes has been corrected successfully. [Please see previous Figure 6, now after revision it became figure 5 on page # 9]
Training image counts adds up to 16340 not 16350.
Dear Reviewer, The image counts corrected to 16340, it was a typo error. [Please see Table # 2, page # 6]
Dataset numbers are very odd you miss out a large number of images without any reasoning.
We used 11,670 images of benign class and 584 images of melanoma. Considering the data of these two classes is imbalanced. Therefore, to handle the class imbalance issue, 4522 melanoma images have been included in ISIC 2019 archive [76]. After that, various data augmentation strategies were performed, including rescaling, width shift, rotation, shear range, horizontal flip, and channel shift, which became 11,670 after augmentation. The reason behind using 11,670 images of benign is to tackle the class imbalance issue. The images of the benign class have been selected arbitrarily from the whole set of images. See sample images in Figure 2 and details of the classes in Table 2.
It isn’t clear but appear no comparison against other networks is performed. The first table seems to just be paper authors scores.
Dear Reviewer, thank you for your valuable suggestion. Table 1 presents the summary of literature, whereas the comparison with state-of-the-art techniques presented in Section 4.2, Table 6.
You say you split and dive the dataset, but you state in section 3.1 you use all ISIC 2020 dataset so the network has learned from the training and validations set.
Dear Reviewer, The whole dataset is not used for experimental purposes, because of the class imbalance issue.
You also need to demonstrate your results on the ISIC leader board to demonstrate your method over others. Not just on your testing set.
The ISIC 2020 test (leader board) results showed that the proposed method obtained 98.04%, which means there was not enough difference between two test accuracies, our test set and the leader board test set, as shown in Table 5. There were 10,982 images in the ISIC 2020 test set on the leader board, with 690 unique patient IDs and 10,292 duplicates. Because the ground truth for ISIC 2020 was not publicly available, the organizer’s statistic on Kaggle area under the Receiver Operating Characteristics Curve (AUC) was used.
The paper need a lot of correcting and checking over and fair comparisons. Statements need to be doubled checked as many do not makes sense or fit in the section.
The whole article is rechecked for fair comparisons, all statements verified wile considering the suggestion of respected reviewer.
Figures need standardising and to be in-house not from google.
Dear Reviewer, we have removed the figure # 1. [Please see page # 2]
Thank you very much for your appreciation and valuable suggestions!!!!!
Reviewer 2 Report
The manuscript has described a useful technique that can improve the accuracy and ease of melanoma diagnosis.
I suggest the following changes:
- An improvement in diagnosis accuracy doesn’t necessarily mean the ability to diagnose earlier, especially if the transfer learning technique still requires dermatoscopy images. Unless the authors can make a compelling argument about how this technique can significantly advance the timing of diagnosis, I would not recommend emphasizing the value of early diagnosis in the article. It can give the readers the incorrect perception that this technique is proven to contribute to early diagnosis.
- In Introduction, the authors stated “American Cancer Society reported that over 100,000 new cases of melanoma would be discovered in 2020, with almost 7,000 patients anticipated to die as a result of cancer.” 2020 was in the past, can the authors cite the actual stats of what happened in 2020? Number of new cases and deaths in 2020 are facts now and no longer predictions.
- In Introduction, the authors stated “According to current figures, a 55% increase in recent skin cancer cases examined annually from 2009 to 2019.” This language is very ambiguous and unclear. Did the number of cases in 2019 increase by 55% compared to the number of cases in 2009? Or is there a 55% increase every year? There also doesn’t seem to be evidence supporting either scenario in the cited paper. Can the authors verify the accuracy of the statistics and clarify the sentence?
- In line 51-53, the authors stated “Previous approaches needed substantial preprocessing, segmentation, and feature extraction operations to categorize skin images.” It seems “previous” should be “former”. Are the authors referring to the dermoscopy approach?
- Many figures are fuzzy, with some words in Fig. 6 not even legible. Can the authors provide high definition figures?
- The bar graph in Fig. 7 is unnecessary as the heights of the bars are essentially the same and not differentiable.
- The article is generally readable, but there are grammatical errors throughout. The language needs substantial improvement to meet publication standards. I recommend hiring a life sciences editor online who is a native speaker.
Author Response
We are thankful to the reviewers for appreciating the strength of our article. We are also thankful for their valuable comments and suggestions which helped us to improve the manuscript (ID: applsci-1713218) entitled “Skin Cancer Disease Detection using Transfer Learning Technique”. The revised version has been prepared to address the reviewers’ suggestions. We have tried our best to answer their questions here and also included the relevant information in the revised paper as well. The author’s responses to the reviewers’ comments are highlighted here in Yellow color whereas the actual modifications necessary to be made in the paper are accentuated in Yellow. Detailed responses to the reviewers’ comments are given below.
Reviewer 2:
- An improvement in diagnosis accuracy doesn’t necessarily mean the ability to diagnose earlier, especially if the transfer learning technique still requires dermatoscopy images. Unless the authors can make a compelling argument about how this technique can significantly advance the timing of diagnosis, I would not recommend emphasizing the value of early diagnosis in the article. It can give the readers the incorrect perception that this technique is proven to contribute to early diagnosis.
We are thankful to the reviewer for this valuable comment the proposed automatic diagnostic system will help the practioners in timely diagnosing the skin cancer.
- In Introduction, the authors stated “American Cancer Society reported that over 100,000 new cases of melanoma would be discovered in 2020, with almost 7,000 patients anticipated to die as a result of cancer.” 2020 was in the past, can the authors cite the actual stats of what happened in 2020? Number of new cases and deaths in 2020 are facts now and no longer predictions.
The latest figures about the number of cases reported in year 2020 and death rate is presented in the article, we also included the statistics regarding the estimated no of cases will be reported in year 2022.
- In Introduction, the authors stated “According to current figures, a 55% increase in recent skin cancer cases examined annually from 2009 to 2019.” This language is very ambiguous and unclear. Did the number of cases in 2019 increase by 55% compared to the number of cases in 2009? Or is there a 55% increase every year? There also doesn’t seem to be evidence supporting either scenario in the cited paper. Can the authors verify the accuracy of the statistics and clarify the sentence?
The statistics regarding the no of cases reported in year 2019 and 2020 updated along with death rate and survival rate.
- In line 51-53, the authors stated “Previous approaches needed substantial preprocessing, segmentation, and feature extraction operations to categorize skin images.” It seems “previous” should be “former”. Are the authors referring to the dermoscopy approach?
Here the previous approaches mean the state-of-the-art techniques.
- Many figures are fuzzy, with some words in Fig. 6 not even legible. Can the authors provide high definition figures?
All the figure updated to accomplish the high-definition figures
- The bar graph in Fig. 7 is unnecessary as the heights of the bars are essentially the same and not differentiable
Response: Dear Reviewer, we have removed the figure 7 as per your suggestion.
- The article is generally readable, but there are grammatical errors throughout. The language needs substantial improvement to meet publication standards. I recommend hiring a life sciences editor online who is a native speaker.
The grammar of the article is rectified thoroughly with the help of a native speaker.
Thank you very much for your appreciation and valuable suggestions!!!!!
Reviewer 3 Report
I suggest to use one term throughout text: dermoscopy or dermatoscopy.
Lines 17-18: There are also other types of skin malignancies.
Line 25: ‘’Skin surface cells’’ is an imprecise term.
Line 26: Melanoma cannot be called benign.
Lines 33-34: Early diagnosis of which type of skin cancer?
Line 44: be ‘’impacted’’ by?
Line 62: abbreviation CNNs to be defined
Line 72: CNNs instead of CNN’s?
Line 384: ‘’benign skin cancer’’ does not exist (benign skin lesions?)
Author Response
We are thankful to the reviewers for appreciating the strength of our article. We are also thankful for their valuable comments and suggestions which helped us to improve the manuscript (ID: applsci-1713218) entitled “Skin Cancer Disease Detection using Transfer Learning Technique”. The revised version has been prepared to address the reviewers’ suggestions. We have tried our best to answer their questions here and also included the relevant information in the revised paper as well. The author’s responses to the reviewers’ comments are highlighted here in YELLOW color whereas the actual modifications necessary to be made in the paper are accentuated in YELLOW. Detailed responses to the reviewers’ comments are given below.
Reviewer 3:
I suggest to use one term throughout text: dermoscopy or dermatoscopy..
Response: Dear Reviewer, used dermoscopy throught the article and replace the dermatoscopy with dermoscopy.
Lines 17-18: There are also other types of skin malignancies.
Response: Dear Reviewer, thank you for your suggestion we have updated the types as per your valuable suggestion.
Line 25: ‘’Skin surface cells’’ is an imprecise term.
Response: Dear Reviewer, we have update the (Squamons Cell Layer) as per your suggestion..
Line 26: Melanoma cannot be called benign.
Response: Dear Reviewer, thank you for valuable comment. We have correct as per your direction.
Lines 33-34: Early diagnosis of which type of skin cancer?
Response: Dear Reviewer, have updated as per your suggestion.
Line 44: be ‘’impacted’’ by?
Response: Dear Reviewer, we have updated as per your suggestion.
Line 62: abbreviation CNNs to be defined
Response: Dear Reviewer, we have write Convolutional Neural Networks (CNNs) as per your suggestion.
Line 72: CNNs instead of CNN’s?
Response: Dear Reviewer, we have replaced CNN’s with CNNs as per your suggestion.
Line 384: ‘’benign skin cancer’’ does not exist (benign skin lesions?)
Response: Dear Reviewer, we have replaced CNN’s with CNNs write the benign skin lessions as per your suggestion.
Thank you very much for your appreciation and valuable suggestions!!!!!